# Origin of Bluetongue Virus Serotype 8 Outbreak in Cyprus, September 2016

**DOI:** 10.3390/v12010096

**Published:** 2020-01-14

**Authors:** Paulina Rajko-Nenow, Vasiliki Christodoulou, William Thurston, Honorata M. Ropiak, Savvas Savva, Hannah Brown, Mehnaz Qureshi, Konstantinos Alvanitopoulos, Simon Gubbins, John Flannery, Carrie Batten

**Affiliations:** 1Pirbright Institute, Woking, Surrey GU24 0NF, UKhannah_brown90@hotmail.co.uk (H.B.); mehnaz.qureshi@pirbright.ac.uk (M.Q.); simon.gubbins@pirbright.ac.uk (S.G.); john.flannery@pirbright.ac.uk (J.F.); carrie.batten@pirbright.ac.uk (C.B.); 2Veterinary Services of Cyprus, Nicosia 1417, Cyprus; vchristodoulou@vs.moa.gov.cy (V.C.); ssavva@vs.moa.gov.cy (S.S.); ackvet@gmail.com (K.A.); 3Met Office, Exeter EX1 3PB, UK; william.thurston@metoffice.gov.uk

**Keywords:** bluetongue virus, BTV-8, reassortment, NGS, midge, NAME

## Abstract

In September 2016, clinical signs, indicative of bluetongue, were observed in sheep in Cyprus. Bluetongue virus serotype 8 (BTV-8) was detected in sheep, indicating the first incursion of this serotype into Cyprus. Following virus propagation, Nextera XT DNA libraries were sequenced on the MiSeq instrument. Full-genome sequences were obtained for five isolates CYP2016/01-05 and the percent of nucleotide sequence (% nt) identity between them ranged from 99.92% to 99.95%, which corresponded to a few (2–5) amino acid changes. Based on the complete coding sequence, the Israeli ISR2008/13 (98.42–98.45%) was recognised as the closest relative to CYP2016/01-05. However, the phylogenetic reconstruction of CYP2016/01-05 revealed that the possibility of reassortment in several segments: 4, 7, 9 and 10. Based on the available sequencing data, the incursion BTV-8 into Cyprus most likely occurred from the neighbouring countries (e.g., Israel, Lebanon, Syria, or Jordan), where multiple BTV serotypes were co-circulating rather than from Europe (e.g., France) where a single BTV-8 serotype was dominant. Supporting this hypothesis, atmospheric dispersion modelling identified wind-transport events during July–September that could have allowed the introduction of BTV-8 infected midges from Lebanon, Syria or Israel coastlines into the Larnaca region of Cyprus.

## 1. Introduction

Bluetongue virus (BTV) is the aetiological agent of bluetongue (BT), a non-contagious disease of domestic and wild ruminants that is transmitted by blood-feeding *Culicoides* midges [1]. The severity of BT varies between species and can cause high mortality, weight loss and disruption of wool growth in sheep, which are the most vulnerable species. In contrast, BT in cattle is usually asymptomatic but, due to the prolonged viremia, cattle play an important role in virus transmission. BT is a notifiable disease by the World Organisation for Animal Health (OIE) and suspicion of disease needs to be reported to government authorities due to its high impact on domestic ruminants.

BTV belongs to the *Orbivirus* genus within the family *Reoviridae* and, as with other members of this genus, it has a linear double-stranded (ds) RNA genome consisting of 10 segments (Seg-1 to Seg-10). The BTV genome encodes for seven structural (VP1 to VP7) and five non-structural proteins (NS1 to NS5) [2,3]. BTV serotype identification primarily relies on the serum neutralisation test, which is based on specific interactions between neutralising antibodies, generated during infection of the host, and the viral VP2 protein located on the outer capsid [4,5]. Based on the serum neutralisation test and/or phylogenetic analysis of VP2, which is the most variable protein, several novel BTV serotypes have been identified [6] in the last decade. Novel BTV serotypes such as BTV-25, -26, and -27 in addition to a number of as-yet unclassified strains belong to a separate cluster on the VP2 phylogenetic tree (the genogroup K clade) and are considered atypical (non-virulent) as they have been solely detected in asymptomatic animals [7,8,9].

BTV was first detected in Cyprus in 1924 and since then several BTV outbreaks have been reported [10] on the island. Earlier outbreaks (1924–1977) involved two serotypes, BTV-3 and BTV-4, while in 1977 over 13% of sheep were affected by BTV-4 alone [10]. Since the 2000s, the global distribution of BTV has changed and several classical BTV serotypes (BTV-1, -2, -3, -4, -6, -8, -9, -11, and -16) have entered European countries leading to widespread disease and economic losses. This shift in the global distribution of BT has been attributed not only to global warming, but also to the identification of indigenous midge species previously not considered as the competent vectors [11]. In Cyprus, the first incursion of BTV-16 was recorded in 2003, and since that time, numerous outbreaks in 2006, 2010, 2011 and 2014 were caused by BTV-4 and/or BTV-16. Presently, the whole territory of Cyprus is classified as an animal movement restriction zone for these two serotypes.

In 2006, BTV-8 emerged in the Netherlands and spread across Belgium, Germany, northern France and Luxemburg. After the winter season, BTV-8 re-emerged and spread to other countries such as Denmark, Switzerland, the Czech Republic and the United Kingdom in 2007. This BTV-8 outbreak had a huge economic impact on the livestock industry in Northern Europe [12] and was the largest outbreak of a single BTV serotype to have occurred to date. A few unique characteristics of this BTV-8 strain were observed, such as the ability to cause serious disease in cattle and goats and infect wild ruminant species [13,14]. In addition, transmission through the transplacental route leading to infection of newborns with or without developmental disorders, and its excretion in semen, contributed to onwards transmission to cattle [15].

Subsequent years of mass vaccination resulted in the eradication of the disease from northern Europe, but BTV-8 re-appeared again in central France during 2015 [16]. Full-genome sequence analysis indicated a high sequence similarity (99.9%) between a French BTV-8 strain isolated in 2008 and in the re-emerged French 2015 BTV-8 strain. This BTV-8 strain is now enzootic in France and it is still considered to pose an economic risk to naïve ruminants [17].

In September 2016, BTV-8 was identified for the first time in Cyprus. The aim of this study was to identify the source of the Cypriot BTV-8 outbreak through full-genome sequencing. The relationship between the Cypriot BTV-8 strain with historic and the most-recent BTV-8 strains circulating in Europe and neighbouring countries was investigated here by two independent approaches: phylogenetic analysis of segment-specific trees and construction of the phylogenetic network.

## 2. Materials and Methods

### 2.1. Epidemiological Investigation

The first three suspected cases of BT in sheep were reported simultaneously on 19 September 2016 by the same private veterinarian in three different locations in Cyprus. Tracings confirmed that these cases were not epidemiologically linked. The observed clinical signs included fever, mouth and lip oedema (swelling), nasal discharge, lameness, abortions and increased mortality. EDTA blood samples from affected ewes were collected and sent to the National Reference Laboratory for Bluetongue virus (NRL-BTV) in Nicosia, Cyprus. The NRL-BTV confirmed the presence of BTV by serogroup-specific real-time RT-PCR on 20 September 2016. EDTA blood samples (*n* = 16) were dispatched to the European Union Reference Laboratory for Bluetongue (EURL-BT) at the Pirbright Institute, UK for serotyping where BTV-8 was confirmed in all submitted samples.

During the BTV-8 outbreak in Cyprus, the farms in Larnaca District were foremost affected and suffered economic losses. An initial epidemiological investigation showed that the clinical signs were observed between 8 and 12 September 2016, indicating that the BTV incursion probably occurred in late August 2016. A sharp increase in mortality was observed in late October 2016 and by 19 December a total of 140 outbreaks had been reported to the OIE [18] comprising 464 deaths in small ruminants (sheep and goats) and 23 cases in cattle without any deaths.

BT control measures were imposed, including the establishment of surveillance and protection zones and animal movement restrictions. In addition, farmers were advised on the use of insecticides and general biosecurity measures with the aim of containing the disease. Moreover, veterinary advice and guidance on the proper supportive care for the ill animals was given in order to reduce the case fatality rate. The last case of BTV-8 was detected in the middle of December 2016. A mass vaccination campaign to vaccinate all sheep, goats and cattle commenced in February 2017 and no further cases of BTV-8 have been reported in Cyprus since.

### 2.2. Clinical Manifestation

Although most of the sheep showed a degree of illness, mortality occurred in pregnant ewes, ewes after lambing and newborns. The most frequently-observed general signs were fever (40.5–42 °C), anorexia, excessive salivation, ulcerative and necrotic lesions of the oral mucosa, hyperaemia and oedema of the conjunctival mucosa, nasal discharge followed by sore muzzle, hyperaemia of the teats and the udder, red rough udder syndrome and agalactia, swollen head and lameness. Two or three of these listed symptoms were found in most of the infected sheep. In addition, an increase in abortion, stillbirth, or delivery of weak or malformed newborn lambs was recorded. The course of the disease varied: in peracute cases, sheep died within 5–7 days of infection while, in chronic cases, some sheep died 2–4 weeks after infection and other sheep were culled due to prolonged morbidity during convalescence. Mild cases which received prompt and proper supportive treatment and care recovered rapidly and completely. However, practitioners later revealed that fertility had been impaired and that milk production had been suboptimal in sheep. In contrast, clinical signs were not observed in cattle and were rare in goats. Cattle and goats were found to harbour the virus (viraemia was confirmed by molecular testing) and animal movement restrictions were put in place.

### 2.3. Diagnostics and Sample Selection for Virus Isolation

#### 2.3.1. The NRL-BTV in Nicosia, Cyprus

The ID Screen Bluetongue Competition ELISA (ID VET, Montpellier, France) was used for the detection of antibodies against the BTV VP7 protein in 992 serum samples (537 sheep; 244 goats; 211 cattle) according to the manufacturer’s instructions. Serogroup-specific real-time RT-PCR assay, targeting the BTV Segment 1 [19], was carried out on a total of 354 EDTA blood samples (275 sheep; 48 goat; 31 cattle). In total, 482 animals were found positive by serological examination and 262 animals were positive using the BTV real-time RT-PCR.

#### 2.3.2. The EURL-BT at the Pirbright Institute, United Kingdom

A single serotype (BTV-8) was identified in all EDTA blood samples (*n* = 16) submitted to the EURL-BT using a serotype specific real-time RT-PCR assay targeting BTV Segment 2 [20]. Washed bloods (*n* = 5) were used as an inoculum for virus propagation and BTV-8 was isolated on KC, Vero and/or BSR cells as described previously [21]. Five BTV-8 isolates, named CYP2016/01 (KC1/BSR1), CYP2016/02 (KC2), CYP2016/03 (KC2), CYP2016/04 (KC2), and CYP2016/05 (KC1/Vero1), were deposited in the Orbivirus Reference Collection at The Pirbright Institute (http://www.reoviridae.org/dsRNA_virus_proteins). These Cypriot isolates (CYP2016/01-05) were later selected for full genome characterisation alongside other BTV isolates from previous submissions: Cyprus (CYP2010/01, CYP2010/02, and CYP2014/01) and Israel (ISR2009/02-03, ISR2009/13-14, ISR2011/04, ISR2010/18, ISR2010/33, and ISR2010/36) (Table 1). In addition, full genome sequencing of the historic BTV reference stains (*n* = 13) was performed and included in the data analysis (Appendix A).

### 2.4. High Throughput Sequencing

Total RNA was extracted from cell culture pellets using TRIzol Reagent (Life Technologies, Paisley, UK) and eluted in 100 µL of nuclease free water (Sigma-aldrich, Gillingham, UK). One microlitre of RNase T1 enzyme was added into each tube and incubated at 37 °C for 30 min in the thermocycler in order to remove ssRNA. DsRNA was purified using the RNA clean and concentrator kit (Zymo, Irvine, CA, USA) according to the manufacturer’s recommendations. The purified dsRNA (8 µL) was denatured by heating at 95 °C for 5 min and the first cDNA strand was synthesised using SuperScript III RT (Life Technologies, Paisley, UK) while the second strand was synthesised using NEBNext (New England BioLabs, Hitchin, UK) according to the manufacturers’ instructions. Double stranded cDNA was purified using the Illustra GFX PCR DNA and Gel Band Purification kit (GE Healthcare, London, UK) and quantified with the Qubit dsDNA HS Assay kit (Life Technologies). The concentration of dscDNA was then adjusted to 0.2 ng µL^−1^ with 10 mM Tris-HCl, pH 8.0 buffer. Libraries were prepared using the Nextera XT library preparation kit and paired end read sequencing was performed using MiSeq Reagent kit v2 (Illumina, San Diego, CA, USA) on the MiSeq benchtop sequencer.

### 2.5. Genome Assembly

A pre-alignment quality check was performed using the FASTQC program v0.11.8 [http://www.bioinformatics.babraham.ac.uk/projects/fastqc/ accessed on 12/10/2019] and the Trim Galore script [22] was used for quality and adapter trimming of FASTQ files along with removal of short sequences (<50 bp). Subsequently, reads were mapped to a range of reference sequences using the BWA-MEM tool [23] and then the DiversiTools software [http://josephhughes.github.io/DiversiTools/accessed on 12/10/2019] was used to generate the consensus sequence. If the consensus sequence contained any gaps, a BLAST search was performed in order to find a more suitable reference sequence. Finally, the consensus sequence was used as a reference sequence to increase the number of BTV reads mapped to the reference, and the final consensus sequence was saved and used for further analysis. Full genome sequencing data of the BTV isolates from Cyprus and Israel were submitted to Genbank (Table 1) along with the historic BTV reference strains (Appendix A).

### 2.6. Phylogenetic Analysis

BTV reference sequences were retrieved from Genbank to represent all BTV genotypes that have been proposed by a BTV genome sequence data resource (BTV-GLUE, http://btv.glue.cvr.ac.uk/#/home). The BTV nomenclature proposed by BTV-GLUE was used throughout this manuscript. For each of the ten BTV segments, a multiple sequence alignment was performed using the Muscle algorithm of the MEGA6 software [24] and phylogenetic trees were reconstructed using IQ-Tree software version 1.3.11.1 [25]. The best-fit model of evolution was selected according to the Bayesian information criterion score calculated using the IQ-Tree software. The maximum likelihood tree was based on the GTR + G4 model of substitution for Segment 1; the GTR + I + G4 model of substitution for Segments 2 and 6; the TIM2 + G4 model of substitution for Segment 3, the TIM2 + I + G2 for Segments 4, 5, 8, 9 and 10; and the TN + I + G4 model of substitution for Segment 7. The reliability of the trees was estimated by ultrafast bootstrap [26] analysis of 1000 replicates and the clade/group/subgroup was supported at a value ≥ 95%. All phylogenetic trees were visualised and rooted on the midpoint in the MEGA6 software.

A multiple alignment of ten coding regions (VP1–VP7 and NS1–NS3) was performed using the Muscle algorithm of MEGA6 software and then FASTA files were concatenated using the SequenceMatrix tool [27] by matching taxon-names. The split network was built using the SplitsTree4 version 4.15.1 software [28] with default settings such as: distances calculated using the UncorrectedP method, a network was calculated using the NeighbourNet method and lastly drawn using the EqualAngle method.

### 2.7. Atmospheric Dispersion Modelling

Simulations were performed with the Numerical Atmospheric-dispersion Modelling Environment (NAME) [29], to estimate the likely source regions of an airborne incursion of BTV-8 into Cyprus. NAME is a Lagrangian particle dispersion model that simulates the release, transport, mixing and removal of material in the atmosphere. NAME calculates the trajectories of a large number of model particles through the time-varying, three-dimensional wind field provided by a Numerical Weather Prediction (NWP) model, with an additional stochastic velocity component included to represent turbulent mixing processes that are unresolved by the driving NWP. NAME has been used on many occasions in the study of the windborne transport of *Culicoides* vectors, both in a post-incursion investigation framework [30,31] and as part of an incursion prediction risk assessment framework [32]. Two types of simulation were performed here: (i) backwards-in-time “air history” simulations, to estimate the origin of air parcels that travelled over Larnaca; and (ii) forwards-in-time “midge dispersal” simulations, to estimate the destination of windborne *Culicoides* midges blown from likely source regions of BTV-8. All NAME simulations used meteorological fields from the operational global configuration of the Met Office NWP model, the Unified Model (UM) [33], which in 2016 provided data on a horizontal grid of approximately 17 km spacing and at a temporal interval of 3 h.

#### 2.7.1. Air History Simulations

Model particles were released continually at a rate of 10 million particles per hour over the Larnaca region, at heights of between 0 and 500 m above ground level (AGL), throughout the period 21 July to 18 September 2016. The trajectory of each model particle was calculated backwards in time from its origin, and at each model timestep the trajectories that were located within the lowest 200 m of the atmosphere were aggregated onto a regular horizontal grid, of approximately 5 km spacing. The resulting trajectory densities represent the near-surface footprint of air parcels that subsequently passed over Larnaca. Three sets of air history simulations were performed, in which the trajectories were terminated after calculating backwards in time for 12, 24 and 36 h, respectively.

#### 2.7.2. Midge Dispersal Simulations

NAME simulations were performed with sources in potential regions of BTV-8, positioned to give complete coverage of the coastline from the Syria–Turkey border to the Gaza–Egypt border. Source 1 covered northern Syria, Source 2 covered southern Syria and northern Lebanon, Source 3 covered central Lebanon, Source 4 covered southern Lebanon and northern Israel, Source 5 covered central Israel, and Source 6 covered southern Israel and Gaza. Simulations were initiated twice a day throughout the period 20 July to 17 September 2016, with model particles being released over the 3-h period around local sunrise and the 3-h period around local sunset, respectively, to represent the diel activity of *Culicoides*. As with the air history simulations, trajectories for each midge dispersal simulation were aggregated onto a 5-km horizontal grid whenever they were within 200 m of the surface, resulting in twice-daily near-surface footprints of midge density from each potential source. Three sets of simulations were performed, with trajectory lifetimes of 12, 24 and 36 h.

## 3. Results

### 3.1. Genogroup, Genotype and Topotype Assignment

BTV isolates sequenced in this study (*n* = 16) were assigned a genotype based on phylogenetic analysis of Segment 2 (Seg-2) (Table 1). Two genotypes such as BTV-8 and BTV-16B were detected in the isolates originating from Cyprus and five different genotypes (BTV-2A, BTV-5, BTV-8, BTV-12, and BTV-24) in the isolates from Israel. In addition, all sequences analysed were classified into eleven genogroups (designated from A to K) and this grouping was statistically supported (Figure 1a). Classical or notifiable BTV serotypes were assigned genogroups A to J, whereas atypical BTV genotypes (represented by the following strains TOV, 379, XJ 1407, V196 XJ 2014, BTV-X ITL2015, and KUW2010/02) were assigned to genogroup K.

For Seg-6, all sequences were assigned to one of the seven genogroups, designated from A to G, and this grouping was statistically supported by the ultrafast bootstrap value ≥ 95, but it did not correspond with the Seg-2 classification. Cypriot 2016/01-05 isolates were assigned to genogroup D along with the Israeli ISR2010/18 isolate (Appendix A).

The majority of the BTV sequences were clearly assigned to an eastern or western topotype based on the phylogenetic analysis of Seg-1, -3, -4, -5, -8, and -9, and this grouping was statistically supported by the ultrafast bootstrap value ≥ 95. Atypical BTV sequences were not classified with either western or eastern viruses possibly indicating their unique origin. Cypriot isolates (CYP2016/01-05) clustered closely with western viruses as did French isolates such as FRA2008/27 and BTV8-15-01 in Seg-1, -3, -4, -5, -8, -9, and -10 (Figure 1 and Appendix A). Unlike for other BTV genome segments, five smaller clusters were statistically distinguished for Seg-7.

### 3.2. Full-Genome Sequencing of Cypriot 2016 Isolates

The percent of nucleotide sequence (%nt) identity between Cypriot isolates (CYP2016/01-05) ranged from 99.92% to 99.95%, and the percent amino acid (%aa) similarity ranged between 99.96% and 99.99%, which corresponded to 2–5 amino acid (aa) changes between isolates. The closest relative to all Cypriot isolates (CYP2016/01-05) was identified as ISR2008/13 (%nt identity 98.42–98.45%; %aa similarity 99.24–99.26%) based on the full genome coding sequence, followed by ISR2010/08 (%nt identity 98.37–98.39%; %aa similarity 99.20–99.22%). The %nt identity between CYP2016/01-05 and French isolates (FRA2008/27, FRA2008/29, and BTV8-15-01) ranged from 98.25% to 98.29% (%aa similarity 99.14–99.19%).

#### 3.2.1. Segment 1

The closest relative to the CYP2016/01 isolate was identified as ISR2008-13 (99.67%) based on the %nt identity, followed by ISR2010-18 (99.57%) and French isolates BTV8-15-01 (99.57%) and FRA2008/27 (99.49%). There was a 1 aa difference between the CYP2016/01 isolate and its closest relative ISR2008-13 at coding position 874. When the CYP2016/01 sequence was compared with the eastern topotype strain CYP2014/01, the %nt identity decreased to 80.10%.

#### 3.2.2. Segment 2

Two BTV-8 isolates, ISR2008/13 (99.46%) and ISR2010/18 (99.46%), were identified as the closest relative for CYP2016/01 based on the %nt identity, but the smallest number of aa changes (*n* = 1) was identified between sequences CYP2016/03 and ISR2010/08. The %nt identity between CYP2016/01 and FRA2008/27 (KP820980) and between CYP2016/01 and BTV-15-01 (KU569991) was 99.22% and 99.29%, resulting in 5 and 4 aa changes, respectively.

#### 3.2.3. Segment 3

Based on the %nt identity, the closest relative to CYP2016/01 was identified as ISR2010-18 (99.75%), followed by ISR2008-13 (99.67%) and BTV8-15-01 (99.35%). No amino acid changes were observed between CYP2016/01 and the Israeli isolates (ISR2008-13 and ISR2010-18), whereas 2 aa changes were recorded at positions 461 and 548 of the VP3 coding region between CYP2016/01 and the French BTV8-15-01.

#### 3.2.4. Segment 4

Isolates CYP2016/01-05 clustered together with several Israeli isolates sequenced in this study such as BTV-2A ISR2010/33, BTV-5 ISR2009/14, BTV-5 ISR2009/13, BTV-12 ISR2010/36, and BTV-5 ISR2011/04 and this grouping was statistically significant (supported by the ultrafast bootstrap value ≥ 95). The %nt identity between CYP2016/01 and other group members ranged from 97.57% (ISR2010-33) to 98.58% (ISR2006/12). In contrast, several BTV-8 isolates from the European countries (NET2007/01, NET2006/04, GRE2008/01, FRA2008/27, BTV8-15-01, BTV-8IT2008, and UKG2007/06) and Israel (ISR2010/18 and ISR2008/13) clearly form a separate subgroup (BTV-8) shown on the phylogenetic tree (Figure 1b). The %nt identity between CYP2016/01 isolate and this BTV-8 cluster ranged from 92.98% (BTV-8IT2008) to 95.31% (NET2007/01).

#### 3.2.5. Segment 5

The closest relatives to CYP2016/01 were identified as ISR2008-13 (99.49%) and ISR2010-18 (99.49%) based on the %nt identity and the Cypriot isolates differed only by 1 aa at position 418 from these isolates. Two and three aa changes were recorded between CYP2016/01 and the French isolates: BTV8-15-01 and FRA2008/27, respectively.

#### 3.2.6. Segment 6

The closest relative to CYP2016/01 was ISR2008/13 (99.82%), followed by 2015 French isolate BTV8-15-01 (99.69%). The CYP2016/01-05 isolates shared 100% similarity with both the Israeli isolates from 2008 and 2010 (ISR2008/13 and ISR2010/18) and French isolates from 2008 (FRA2008/27) and 2015 (BTV8-15-01).

#### 3.2.7. Segment 7

Based on the %nt identity, the closest relatives to CYP2016/01 were BTV-24 ISR2009-02 (99.82%) and BTV-24 ISR2009-03 (99.82%), followed by BTV-24 ISR2008-2 (99.74%), and all three sequences shared 100% similarity on the aa level and clustered together into a statistically supported subgroup (Figure 1d). BTV-8 sequences clustered together into a statistically significant subgroup (labelled as BTV-8 in Figure 1c), the %nt identity between CYP2016/01 and the BTV-8 sequences in this subgroup ranged from 98.12% (ISR2010-08) to 98.42% (UKG2007/06).

#### 3.2.8. Segment 8

For Seg-8, two isolates; ISR2010-18 (99.73%) and FRA2008/27 (99.73%) were identified as the closest relatives to CYP2016/01 isolate. CYP2016/04 differed by 1 aa at the coding position 216 of the NS2 protein in comparison to the remaining CYP2016 isolates (01, 02, 03, and 05).

#### 3.2.9. Segment 9

For Seg-9, the closest relative to CYP2016/01 was identified as BTV-15 ISR2006/11 (99.05%) and 3 aa changes were identified between both sequences; these sequences also clustered into a statistically supported subgroup on the phylogenetic tree (Figure 1d). In contrast, BTV-8 isolates from European outbreaks (FRA2008/07, BTV8-15-01, NET2006/04, NET2007/01, BTV-8IT2008, UKG2007/06 and GRE2008/01) and BTV-8 from Israeli outbreaks (ISR2010-18 and ISR2008-13) formed a distinct subgroup (labelled as BTV-8) on the phylogenetic. The %nt identity between CYP2016/01 and the BTV-8 sequences in the BTV-8 subgroup ranged from 86.75% (BTV-8 IT2008) to 90.33% (FRA2008/27 and ISR2008/13). The location change of the Cypriot isolates (CYP2016/01-06) on the phylogenetic tree in comparison to other BTV-8 isolates indicates a high probability of a reassortment event.

#### 3.2.10. Segment 10

Based on the %nt identity, the closest relative to CYP2016/01 was identified as BTV-5 ISR2009/13 (98.88%) and both these sequences clustered closely with BTV-5 ISR2009/14 and BTV-15 ISR2006/11 (KP821098) (Figure 1e). The %nt identity between CYP2016/01 and other BTV-8 sequences ranged from 92.50% (BTV-8IT2008) to 95.27% (NET2006/04, NET2007/01, and ISR2010/18).

### 3.3. Phylogenetic Network

The split network was estimated from an alignment of 18,621 characters for each of 73 taxa, which sequence composed of the concatenated coding regions of BTV (VP1–VP7 and NS1–NS3). The Neighbor-Net network shows the split between the BTV strains belonging to the western and eastern topotype in addition to the formation of separate genogroup K containing the atypical BTV strains (Figure 2). The CYP2016/01-05 sequences clustered together with other western topotype strains belonging to genogroup D, but clearly split from the European BTV-8 strains from the 2006–2008 and 2015 outbreaks as well as the BTV-8 vaccine and reference strains (Figure 2b).

### 3.4. Atmospheric Dispersion Modelling

Aggregated NAME air history maps (Figure 3) show that the most common origin of air that reached the Larnaca region over the period under consideration was between the west and north west of Cyprus, around the area of Crete and Rhodes. However, against this background of a dominant westerly origin of air reaching Larnaca, there is also an eastward extension of the air history towards the coastlines of Syria, Lebanon and Israel. Some of the air that reached Larnaca within 24 h had origins along the central Lebanon coastline, and potentially from coasts of northern Israel and central Syria. Extending the timeframe to consider air that took up to 36 h to reach Larnaca, the potential area of origin can be extended to include most of the coastline of Syria and Israel. None of the 12-h air histories extended far enough east to reach any of these coastlines (not shown).

The NAME midge dispersal simulations are summarised as a list of potential incursions in Table 2, constructed from manual inspection of the aggregated trajectory densities from each source in each of the twice-daily simulations, examples of which are shown in Appendix A. Potential incursions with a 24-h flight time would have likely only come from between southern Syria and southern central Lebanon. Extending flight duration to 36 h, the potential sources that covered the coastline from the Syria–Turkey border to central Israel could have be responsible for an incursion into the Larnaca region of Cyprus. It is unlikely that the source in southern Israel or Gaza was responsible for any incursions. None of the 12-h midge dispersal simulation sources had potential incursions into the Larnaca region.

## 4. Discussion

In this study, we aimed to investigate the source of the BTV-8 which emerged in Cyprus in September 2016 through full genome sequencing. Based on the Segment-2 sequencing data alone, the BTV-8 CYP2016/01-05 had a high %nt identity with BTV-8 strains from Israel (ISR2008/13 and ISR2010/18) as well as BTV-8 strains from France (FRA2008/27 and BTV-15-01). However, the phylogenetic trees constructed for the individual segments of BTV provide evidence of reassortment occurring in individual Segments 4, 7, 9 and 10. The closest relative for CYP2016/01-05 in these Segments 4, 7, 9, and 10 were BTV-4A ISR2006/12, BTV-24 ISR2009/02, BTV-15 ISR2006/11 and BTV-5 ISR2009/13, respectively. Since the closest relative for each of ten BTV segments originated from Israel, this provides enough evidence to rule out the importation of BTV-8 infected animals from central Europe (e.g., France) as a potential source of the BTV-8 outbreak in Cyprus. It is highly likely that the Cypriot BTV-8 strain emerged in an area where multiple BTV strains were co-circulating allowing for the exchange of segments.

It has been shown previously that reassortment is a common evolutionary process and that genetic exchange of any of ten segments is equally likely to happen [34]. In the geographical locations where multiple distinct BTV serotypes are co-circulating, the opportunities for reassortment are higher and the emergence of novel reassortants, possibly of altered virulence, was previously reported from countries such as India and Israel [35,36]. In Israel, a novel and more-virulent BTV-6 reassortant has recently been spreading throughout the Golan Heights, causing classical BTV signs and death in sheep and cattle [35]. In India, a single variant of Segment 5, which was probably introduced to India as part of an exotic western serotype, recombined with other BTV serotypes (including BTV-5, BTV-9, BTV-12, and BTV-16) and become the most dominant variant in India since 2008 [36], highlighting its fitness advantage over other Segment 5 variants.

Countries such as Israel, Lebanon, Syria and Jordan are considered the most likely donor of new BTV serotypes to Cyprus based on the available epidemiological data (reported circulation of BTV-8) and their proximity to the island. In Turkey, BTV-8 has been notified for the first time as a new serotype during the outbreak in 2018, but not earlier [37]. In Israel, multiple BTV serotypes (BTV-2, BTV-4, BTV-5, BTV-6, BTV-8, BTV-12, BTV-15, BTV-16, and BTV-24) have been co-circulating in sheep and cattle since 2006 [35,38,39]. BTV-8 (ISR2008/13) was first reported in Israel in 2008 and it is believed that its entrance into Israel occurred through the importation of infected animals from northern Europe [40]. Similarly, as occurred during the European BTV-8 epizootic (2006–2010), in Israel the BTV-8 strain (ISR2008/13) was associated with clinical signs of infection in cattle [40]. In contrast, no clinical signs were reported in cattle during the 2016 BTV-8 outbreak in Cyprus, indicating the possibility that the different genome constellation of BTV-8 CYP2016 affects pathogenicity in cattle.

BTV-8 was detected in 2009 and 2010 in Israel and then in 2015 [38] along with a number of other BTV serotypes, which would have provided the opportunity for reassortment. In Israel, co-circulation of several BTV serotypes on the same farm has been reported previously [41], and mixed infections with two or three different BTV serotypes were identified on several occasions [38]. Therefore, it is likely that the most recent BTV-8 circulating in Israel underwent reassortment over time and now differs from its ancestor, BTV-8 ISR2008/13. Although there are limited data on the presence of BT in Jordan and Syria, due to lack of systematic surveillance, it is likely that several BTV serotypes are also co-circulating in these countries. In Lebanon, six BTV serotypes (BTV-1, BTV-4, BTV-6, BTV-8, BTV-16 and BTV-24) were reported in 2011 with mixed-serotype infections reported in the same animals [42]. Unfortunately, no full-genome sequencing data of the BTV-8 Lebanon strain are available.

Two routes of BT incursion into Cyprus are possible, either through importation of BTV-8 positive animals or via long-distance wind dispersion (LDWD) of *Culicoides* vectors. Considering that first BT cases occurred in three different locations in Cyprus that were not epidemiologically linked, the importation of BTV-8 through animal movement is less likely to have happened. Historical meteorological data indicate that the dry air that reached the Larnaca district between 21 July and 18 September 2016 commonly came from the west and north west of Cyprus, which is consistent with climatological summer airflow in the region [43]. Nevertheless, the “air history” simulations support the hypothesis that BTV-8 infected midges could be blown across the Mediterranean Sea from the central Lebanon coastline to the Larnaca district. Moreover, several events of potential midge incursion were identified between 1 August and 11 September 2016 using the NAME midge dispersal simulations. The most likely incursion event based on all the methodological evidence was identified on 7/8 September 2016. This may be slightly too-late to match the incubation period of 5–10 days for BT; however, it cannot be discounted that the BTV-8 outbreak in Cyprus occurred as a consequence of two earlier incursion events (on 20/21 and 29/30 August 2016). In the past, the LDWD of midges was proposed as a probable pathway for the introduction of African horse sickness [44] in 1960 and BT [45] in 1977 into Cyprus from Turkey and Syria/Turkey, respectively. These incursions were later reanalysed using the Tool for Assessing Pest and Pathogen Airborne Spread (TAPPAS) [46], which also suggested both incidents of LDWD were plausible.

Due to frequent reassortment, the evolutionary history of complete BTV genomes cannot be represented by a single phylogenetic tree. This is similar for other viruses, plants or bacteria, which utilise recombination/hybridisation and gene transfer as common evolutionary process. The bifurcating tree assumes that, once two strains diverge, they do not interact with one another in the future. Such an assumption is incorrect for BTV as the exchange of segments between different strains occurs frequently during co-infection of the host. To accommodate for this feature of BTV evolution, the use of a phylogenetic network was investigated to provide a simple visualisation of the epidemiological relationship between strains and display the patterns which a tree representation might overlook. Several different methods to construct the phylogenetic network from sequence alignments are available including split decomposition, Neighbor-net, parsimony split or median networks. Here, we used the Neighbor-Net distance-based method as it allows for the generation of a network comprising a hundred taxa which can be scaled up if required. Although the construction of ten individual phylogenetic trees is an accurate way to analyse the sequencing data, it is also time consuming and maybe unwieldy to present the data to non-scientific audiences during disease outbreaks. In the constructed network, the spilt between the CYP2016/01-05 isolates and other BTV-8 isolates from 2006–2008 (France, the Netherlands Israel, Italy, Greece, and the United Kingdom) and 2015 (France) was clearly observed. This is in-line with the phylogenetic analysis of individual BTV segments supporting the hypothesis that there is no direct epidemiological link between the Cypriot BTV-8 and the northern/central Europe BTV-8 strains. Considering that the %nt identity between Cypriot isolates (99.92–99.95%) was only slightly greater than the %nt identity between Cypriot isolates and their closest relative (ISR2008/13), 98.42–98.45%, the split network constructed here provided an accurate representation of strains originating from separate outbreaks. Although the phylogenetic network is not as common as a tree to display the evolutionary relationship between taxa, it was successfully used for tracing human-to-human transmission during the 2013–2015 Ebola epidemic in Sierra Leone [47], for supporting the field epidemiological investigation during the highly pathogenic avian influenza outbreak in British Columbia [48] or for displaying the geographical relationship amongst the serotypes of the Palyam serogroup of the *Orbivirus* genus [49]. In the absence of other investigative techniques such as meteorological modelling, the use of phylogenetic networks can be used to determine the origins of BTV strains.

## 5. Conclusions

In this study, we demonstrated that the full genome sequencing data can be successfully used in tracing the source of BT outbreaks. The phylogenetic analysis of all individual BTV segments provides more valuable genetic information than the Segment 2 sequence alone. The nt% between the Cypriot isolates originating from a single outbreak was 99.9%. Slightly lower %nt (98.2–98.4%) was observed between the Cypriot isolates and other BTV-8 isolates from 2006–2008 (France, the Netherlands Israel, Italy, Greece, and the United Kingdom) and 2015 (France). However, based on the phylogenetic analysis of individual trees and a phylogenetic network analysis presented here, Cyprus isolates are not directly linked to these outbreaks, and represent a separate outbreak/new incursion rather than continuation of BTV-8 spread from Europe.

The analysis of ten separate phylogenetic trees may be impractical to present to non-scientific audiences during BT outbreak. We demonstrated that a single phylogenetic network can be used to visualise evolutionary relationships between the whole genome sequences of different BTV strains (e.g., belonging to the same serotype). In the constructed network, the spilt between the CYP2016/01-05 isolates and other BTV-8 isolates from 2006–2008 (France, the Netherlands Israel, Italy, Greece, and the United Kingdom) and 2015 (France) was clearly observed indicating that ~98% nt can be sufficient to distinguish between unlinked outbreaks of single serotype. Although the closest relative of BTV-8 CYP2016 was identified as the Israeli ISR2008/13 isolate, it is more likely, based on all the meteorological evidence, that the BTV-8 incursion into the Larnaca district occurred via long-distance wind dispersion of infected midges blown across the Mediterranean Sea from the Lebanon coastline, rather than Israel.

## Figures and Tables

**Figure 1 viruses-12-00096-f001:**
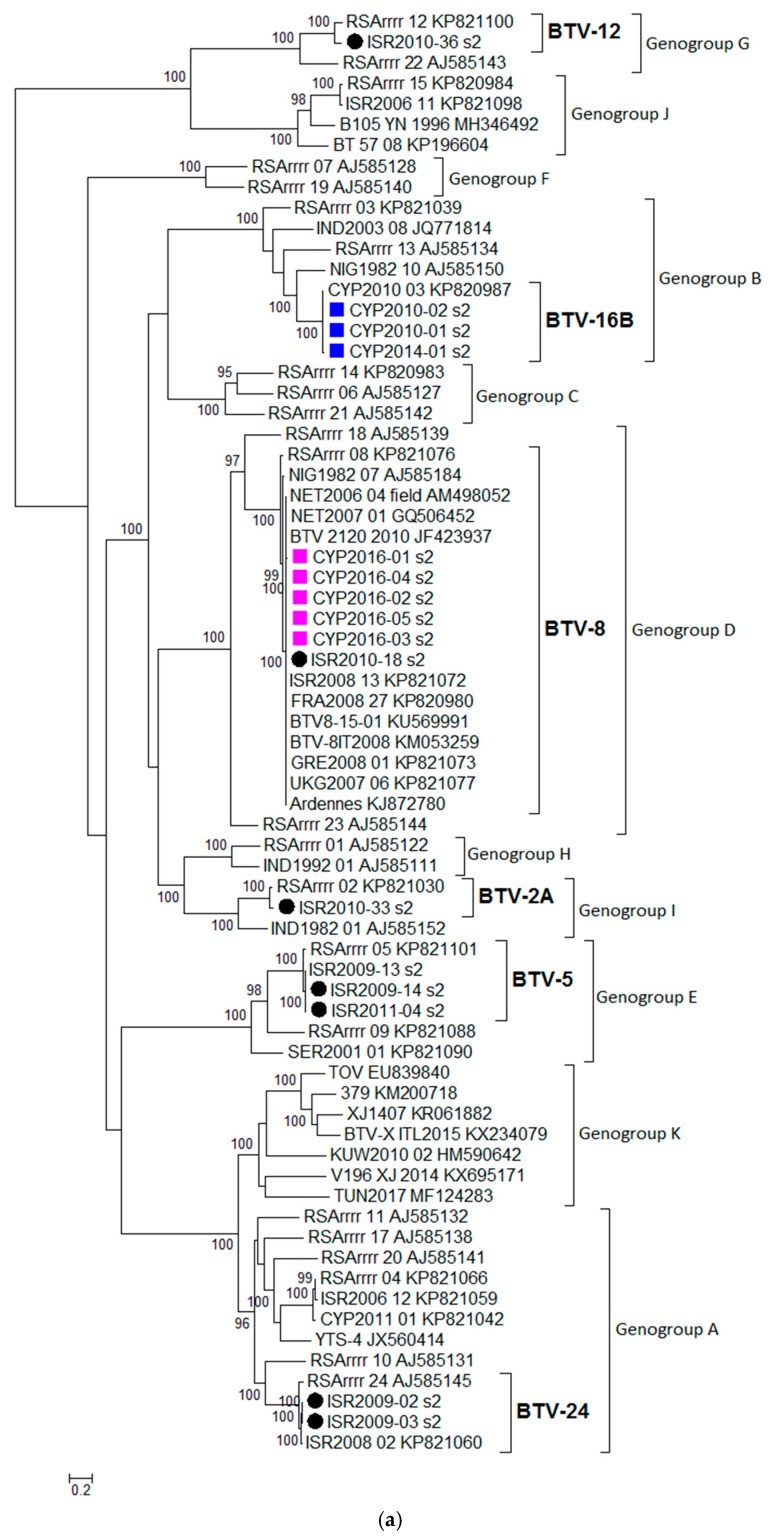
Phylogenetic trees were constructed for the coding regions of BTV: (**a**) VP2 protein (Segment 2); (**b**) VP4 protein (Segment 4); (**c**) VP7 protein (Segment 7); (**d**) VP6 protein (Segment 9); and (**e**) NS3 protein (Segment 10). Maximum likelihood trees were constructed using IQ-Tree software [25] and the reliability of each tree was estimated by ultrafast bootstrap [26] analysis of 1000 replicates (bootstrap values of < 95% are not displayed).

**Figure 2 viruses-12-00096-f002:**
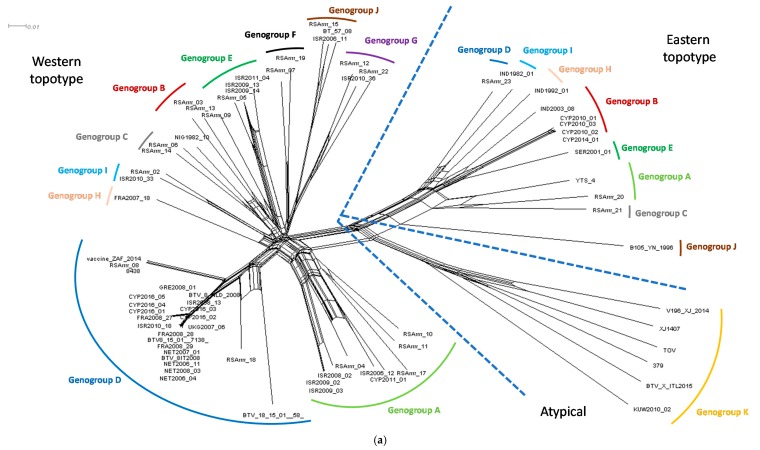
(**a**) The Neighbor-Net network was estimated from an alignment of 18,621 characters for each of 73 taxa, which sequence composed of the concatenated coding regions of BTV (VP1–VP7 and NS1–NS3); and (**b**) genogroup D of the Neighbor-Net network. Nomenclature used in-line with that proposed by the BTV-GLUE resource for the BTV Segment 2. Solid coloured lines were added manually to indicate the genogroup while the dashed blue lines were added manually to indicate the distinct differences between the western and eastern topotypes, and genogroup K which contains the atypical BTV strains.

**Figure 3 viruses-12-00096-f003:**
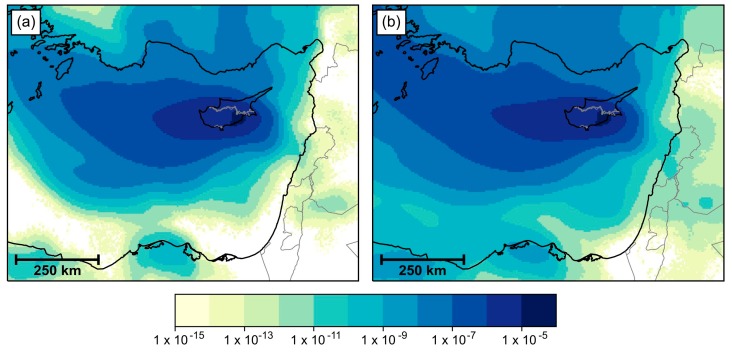
Maps of the near-surface history of air passing over the Larnaca region, aggregated over the period 21 July to 18 September 2016. Air parcels with a lifetime of (**a**) 24 h and (**b**) 36 h are shown.

**Table 1 viruses-12-00096-t001:** BTV isolates sequenced during this study and associated Genbank accession number for the genome segments.

Isolate ^1^	Genotype	Passage Sequenced	Seg-1 (VP1)	Seg-2 (VP2)	Seg-3 (VP3)	Seg-4 (VP4)	Seg-5 (NS1)	Seg-6 (VP5)	Seg-7 (VP7)	Seg-8 (NS2)	Seg-9 (VP6)	Seg-10 (NS3)
CYP2016/01	BTV-8	KC1/BSR1/KC1	MN710077	MN710193	MN710106	MN710135	MN710164	MN710222	MN710251	MN710280	MN710309	MN710338
CYP2016/02	BTV-8	KC3/BSR1	MN710078	MN710194	MN710107	MN710136	MN710165	MN710223	MN710252	MN710281	MN710310	MN710339
CYP2016/03	BTV-8	KC3	MN710079	MN710195	MN710108	MN710137	MN710166	MN710224	MN710253	MN710282	MN710311	MN710340
CYP2016/04	BTV-8	KC3	MN710080	MN710196	MN710109	MN710138	MN710167	MN710225	MN710254	MN710283	MN710312	MN710341
CYP2016/05	BTV-8	KC1/Vero1/KC3	MN710081	MN710197	MN710110	MN710139	MN710168	MN710226	MN710255	MN710284	MN710313	MN710342
CYP2010/01	BTV-16B	KC2	MN710082	MN710198	MN710111	MN710140	MN710169	MN710227	MN710256	MN710285	MN710314	MN710343
CYP2010/02	BTV-16B	KC3	MN710083	MN710199	MN710112	MN710141	MN710170	MN710228	MN710257	MN710286	MN710315	MN710344
CYP2014/01	BTV-16B	KC4	MN710084	MN710200	MN710113	MN710142	MN710171	MN710229	MN710258	MN710287	MN710316	MN710345
ISR2009/02	BTV-24	KC3	MN710085	MN710201	MN710114	MN710143	MN710172	MN710230	MN710259	MN710288	MN710317	MN710346
ISR2009/03	BTV-24	KC3	MN710086	MN710202	MN710115	MN710144	MN710173	MN710231	MN710260	MN710289	MN710318	MN710347
ISR2009/13	BTV-5	E1/BHK2/KC1	MN710087	MN710203	MN710116	MN710145	MN710174	MN710232	MN710261	MN710290	MN710319	MN710348
ISR2009/14	BTV-5	KC4	MN710088	MN710204	MN710117	MN710146	MN710175	MN710233	MN710262	MN710291	MN710320	MN710349
ISR2010/18	BTV-8	E1/V1/KC3	MN710089	MN710205	MN710118	MN710147	MN710176	MN710234	MN710263	MN710292	MN710321	MN710350
ISR2011/04	BTV-5	KC1	MN710090	MN710206	MN710119	MN710148	MN710177	MN710235	MN710264	MN710293	MN710322	MN710351
ISR2010/33	BTV-2A	KC1/BHK1/KC2	MN710091	MN710207	MN710120	MN710149	MN710178	MN710236	MN710265	MN710294	MN710323	MN710352
ISR2010/36	BTV-12	KC1/BHK1/KC2	MN710092	MN710208	MN710121	MN710150	MN710179	MN710237	MN710266	MN710295	MN710324	MN710353

^1^ Isolate name, CYP = Cyprus, ISR = Israel, followed by year of specimen collection.

**Table 2 viruses-12-00096-t002:** Date and time of NAME midge dispersal simulations that suggested potential incursions into the Larnaca region from Syria, Lebanon or Israel.

Event	Date and Time	24-h Midge Sources ^1^	36-h Midge Sources ^1^
(a)	11 September 2016 SS	3	3
	11 September 2016 SR		4
(b)	08 September 2016 SR		2
	07 September 2016 SS ^2^	2, 3	2, 3
	07 September 2016 SR		2, 3
(c)	30 August 2016 SR ^2^		1, 2, 3
	29 August 2016 SS	2, 3	2, 3
	29 August 2016 SR		2, 3, 4
(d)	24 August 2016 SR	None identified	3, 4
(e)	23 August 2016 SR	None identified	1, 2, 3, 4
(f)	21 August 2016 SR		3
	20 August 2016 SS ^2^	2, 3	2, 3
	30 August 2016 SR		2, 3, 4
	29 August 2016 SS		3
	19 August 2016 SR		3, 4, 5
(g)	01 August 2016 SS	2, 3	2, 3
	01 August 2016 SR		1, 2, 3
	31 July 2016 SS		1
	31 July 2016 SR		1, 2, 3, 4
	30 July 2016 SS		3

^1^ Sources: (1) northern Syria; (2) southern Syria and northern Lebanon; (3) central Lebanon; (4) southern Lebanon and northern Israel; (5) central Israel; and (6) southern Israel and Gaza; ^2^ Further details of this potential midge incursions are presented in Appendix A.

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
