# Peer review of "Origin of Bluetongue Virus Serotype 8 Outbreak in Cyprus, September 2016"

_viruses, 2020, doi:10.3390/v12010096_

Round 1
Reviewer 1 Report
This paper is a very worthwhile analysis. The discussion of phylogenetics is strong and sensible but I needed to read through the work twice-so I think it might be presented a bit more clearly. Examination of the individual trees in context with the author's interpretation of the epidemiology is well supported. It is, as the authors themselves note, cumbersome to examine and try to interpret all of the trees individually. The "Neighbor-Net network" and interpretation of its topography was not well introduced or explained, but perhaps this is well known? A bit more of an introduction to this method would be helpful for readers who are unfamiliar, and is needed to support the conclusion that this may be a more effective way (than...?) to visualize and communicate complicated phylogenies. Although the distinctions between the "topotypes" is clear, I'm not sureI also want to know why the lines separating topotypes is so wavy/noisy? The crux of the method is to understand the originsof this outbreak. Please rigorously address how much is known about the genotypes circulating in the region. Also, please be transparent in the discussion that your conclusion is based on very small % differences in sequences, and how the phylogenies support this nonetheless. Some discussion of intrinsic rate of evolution might also be helpful, since we are looking at sequences from many timepoints spread over a long period of time. I don't have any expertise to predict how temporal structure in these sequences might affect their perceived relationships. One thing that would have been helpful, especially in re the discussion of wind and midge dispersal etc., is a more detailed map that clearly indicates the wind simulations and midge incursions you are proposing.Author Response
Reviewer-1
This paper is a very worthwhile analysis. The discussion of phylogenetics is strong and sensible but I needed to read through the work twice-so I think it might be presented a bit more clearly. Examination of the individual trees in context with the author's interpretation of the epidemiology is well supported. It is, as the authors themselves note, cumbersome to examine and try to interpret all of the trees individually.
The "Neighbor-Net network" and interpretation of its topography was not well introduced or explained, but perhaps this is well known?
We thank the reviewer for their appreciation of our work and that they too agree with our decision to avoid presenting multiple phylogenetic trees. There are a number of publications and books available concerning the phylogenetic network analysis, so we did not consider it appropriate to provide a full explanation of the technique. Therefore, in this study, we investigated if the use of this tool (the phylogenetic network) can provide easy visualisation of full-genome sequencing data of BTV.
A bit more of an introduction to this method would be helpful for readers who are unfamiliar, and is needed to support the conclusion that this may be a more effective way (than...?) to visualize and communicate complicated phylogenies.
Line 77-79 Text changed from “In September 2016, BTV-8 was identified for the first time in Cyprus. The aim of this study was to identify the source of the Cypriot BTV-8 outbreak through full-genome sequencing. Ten individual viral segments of the Cypriot BTV-8 strain were compared with historic and the most-recent BTV-8 strains circulating in Europe and neighbouring countries.”
Line 77-79 Text changed to “In September 2016, BTV-8 was identified for the first time in Cyprus. The aim of this study was to identify the source of the Cypriot BTV-8 outbreak through full-genome sequencing. The relationship between the Cypriot BTV-8 strain with historic and the most-recent BTV-8 strains circulating in Europe and neighbouring countries was investigated here by two independent approaches: phylogenetic analysis of segment-specific trees and construction of the phylogenetic network. “
Line 524-549 Text changed from “Due to frequent reassortment, the evolutionary history of BTV cannot be represented by a single phylogenetic tree. In this study, we demonstrated that a phylogenetic network of BTV, utilising full-genome sequencing data, provides a simple visualization of the epidemiological relationship between strains. The Neighbor-Net distance-based method used here allows for the generation of a network for nearly a hundred taxa which can be scaled up if required. Although the construction of ten individual phylogenetic trees is an accurate way to analyse the sequencing data it is also time consuming and maybe unwieldy to present the data to non-scientific audiences during disease outbreaks. In the constructed network, the spilt between the CYP2016/01-05 isolates and other BTV-8 isolates from 2006-2008 (France, the Netherlands Israel, Italy, Greece, and the United Kingdom) and 2015 (France) was clearly observed. This is in-line with the phylogenetic analysis of individual BTV segments supporting the hypothesis that there is no direct epidemiological link between the Cypriot BTV-8 and the northern/central Europe BTV-8 strains. Considering that the %nt identity between Cypriot isolates (99.92% to 99.95%) was only slightly greater than the %nt identity between Cypriot isolates and their closest relative (ISR2008/13) 98.42-98.45%, the split network constructed here provided an accurate representation of these strains. Although the phylogenetic network is not as common as a tree to display the evolutionary relationship between taxa, it has been successfully used for tracing human-to-human transmission during the 2013-2015 Ebola epidemic in Sierra Leone [44], for supporting the field epidemiological investigation during the highly pathogenic avian influenza outbreak in British Columbia [45] or for displaying the geographical relationship amongst the serotypes of the Palyam serogroup of the Orbivirus genus [46]. In the absence of other investigative techniques such as meteorological modelling, the use of phylogenetic networks can be used to determine the origins of BTV strains.”
Line 524-549 Text changed to “Due to frequent reassortment, the evolutionary history of complete BTV genomes cannot be represented by a single phylogenetic tree. This is similar for other viruses, plants or bacteria, which utilize recombination/hybridization and gene transfer as common evolutionary process. The bifurcating tree assumes that once two strains diverge, they do not interact with one another in the future. Such an assumption is incorrect for BTV as the exchange of segments between different strains occurs frequently during co-infection of the host. To accommodate for this feature of BTV evolution, the use of a phylogenetic network was investigated to provide a simple visualization of the epidemiological relationship between strains and display the patterns which a tree representation might overlook.
Several different methods to construct the phylogenetic network from sequence alignments are available including split decomposition, Neighbor-net, parsimony split or median networks. Here, we used the Neighbor-Net distance-based method as it allows for the generation of a network comprising a hundred taxa which can be scaled up if required. Although the construction of ten individual phylogenetic trees is an accurate way to analyse the sequencing data, it is also time consuming and maybe unwieldy to present the data to non-scientific audiences during disease outbreaks. In the constructed network, the spilt between the CYP2016/01-05 isolates and other BTV-8 isolates from 2006-2008 (France, the Netherlands Israel, Italy, Greece, and the United Kingdom) and 2015 (France) was clearly observed. This is in-line with the phylogenetic analysis of individual BTV segments supporting the hypothesis that there is no direct epidemiological link between the Cypriot BTV-8 and the northern/central Europe BTV-8 strains. Considering that the %nt identity between Cypriot isolates (99.92% to 99.95%) was only slightly greater than the %nt identity between Cypriot isolates and their closest relative (ISR2008/13) 98.42-98.45%, the split network constructed here provided an accurate representation of strains originating from separate outbreaks.”
Although the distinctions between the "topotypes" is clear, I'm not sure I also want to know why the lines separating topotypes is so wavy/noisy?
Figure 2. Lines separating eastern, western and atypical BTV strains were corrected.
Figure 2 description was changed for clarity to “Figure 2. a) The Neighbor-Net network was estimated from an alignment of 18621 characters for each of 73 taxa, which sequence composed of the concatenated coding regions of BTV (VP1-VP7, NS1-NS3); b) genogroup D of the Neighbor-Net network. Nomenclature used in-line with that proposed by the BTV-GLUE resource for the BTV segment 2. Solid coloured lines were added manually to indicate the genogroup while the dashed blue line were added manually to indicate the distinct differences between the western and eastern topotypes, and genogroup K which contains the atypical BTV strains.”
The crux of the method is to understand the origins of this outbreak. Please rigorously address how much is known about the genotypes circulating in the region.
We have found as many information as possible on the circulating BTV in Cyprus and in the neighbouring countries and have included them in the introduction or discussion section. We have provided further focus on the BTV-8 reports as this was the main topic of the study. This is clearly supported in the manuscript as indicated below (with the inclusion of relevant changes):
Line 52-54: “BTV was first detected in Cyprus in 1924 and since then a number of BTV outbreaks have been reported [10] on the island. Earlier outbreaks (1924-1977) involved two serotypes BTV-3 and BTV-4 while in 1977 over 13% of sheep were affected by BTV-4 alone [10].”
Line 76: “In September 2016, BTV-8 was identified for the first time in Cyprus. The aim of this study was to identify the source of the Cypriot BTV-8 outbreak through full-genome sequencing.”
Line 490 Text changed from “Countries such as Israel, Lebanon, Syria and Jordan are considered the most likely donor of new BTV serotypes to Cyprus based on the available epidemiological data and their proximity to the island.”
Line 490 Text changed to “Countries such as Israel, Lebanon, Syria and Jordan are considered the most likely donor of new BTV serotypes to Cyprus based on the available epidemiological data (reported circulation of BTV-8) and their proximity to the island.”
Line 493 Text added “In Turkey, BTV-8 has been notified for the first time as a new serotype during the outbreak in 2018, but not earlier [37].
Line 506-511: “Although there are limited data on the presence of BT in Jordan and Syria, due to lack of systematic surveillance, it is likely that several BTV serotypes are also co-circulating in these countries. In Lebanon, six BTV serotypes (BTV-1, BTV-4, BTV-6, BTV-8, BTV-16 and BTV-24) were reported in 2011 with mixed-serotype infections reported in the same animals [41]. Unfortunately, no full-genome sequencing data of the BTV-8 Lebanon strain is available.”
Also, please be transparent in the discussion that your conclusion is based on very small % differences in sequences, and how the phylogenies support this nonetheless.
Lines 558-578: text changed from “In this study, we have demonstrated that the full genome sequencing data can be successfully used in tracing the source of BT outbreaks. The phylogenetic analysis of all individual BTV segments provides more valuable genetic information than the segment 2 sequence alone. However, the analysis of ten separate phylogenetic trees may be impractical to present to non-scientific audience during BT outbreak. We have demonstrated that a single phylogenetic network can be used to visualise evolutionary relationships between the whole genome sequences of different BTV strains (e.g. belonging to the same serotype). In the constructed network, the spilt between the CYP2016/01-05 isolates and other BTV-8 isolates from 2006-2008 (France, the Netherlands Israel, Italy, Greece, the United Kingdom) and 2015 (France) was clearly observed despite the close %nt between both groups. Although the closest relative of BTV-8 CYP2016 was identified as the Israeli ISR2008/13 isolate, it is more likely, based on all the meteorological evidence that the BTV-8 incursion into the Larnaca district occurred via long-distance wind dispersion of infected midges blown across the Mediterranean Sea from the Lebanon coastline, rather than Israel.”
Line 558-578 was changed to “In this study, we have demonstrated that the full genome sequencing data can be successfully used in tracing the source of BT outbreaks. The phylogenetic analysis of all individual BTV segments provides more valuable genetic information than the segment 2 sequence alone. The nt% between the Cypriot isolates originating from a single outbreak was 99.9%. Slightly lower %nt (98.2% to 98.4%) was observed between the Cypriot isolates and other BTV-8 isolates from 2006-2008 (France, the Netherlands Israel, Italy, Greece, the United Kingdom) and 2015 (France). However, based on the phylogenetic analysis of individual trees and a phylogenetic network analysis presented here, Cyprus isolates are not directly linked to these outbreaks, and represent a separate outbreak/new incursion rather than continuation of BTV-8 spread from Europe.
The analysis of ten separate phylogenetic trees may be impractical to present to non-scientific audience during BT outbreak. We have demonstrated that a single phylogenetic network can be used to visualise evolutionary relationships between the whole genome sequences of different BTV strains (e.g. belonging to the same serotype). In the constructed network, the spilt between the CYP2016/01-05 isolates and other BTV-8 isolates from 2006-2008 (France, the Netherlands Israel, Italy, Greece, the United Kingdom) and 2015 (France) was clearly observed indicating that ~98% nt can be enough to distinguish between unlinked outbreaks of a single BTV serotype. Although the closest relative of BTV-8 CYP2016 was identified as the Israeli ISR2008/13 isolate, it is more likely, based on all the meteorological evidence that the BTV-8 incursion into the Larnaca district occurred via long-distance wind dispersion of infected midges blown across the Mediterranean Sea from the Lebanon coastline, rather than Israel.”
Some discussion of intrinsic rate of evolution might also be helpful, since we are looking at sequences from many timepoints spread over a long period of time. I don't have any expertise to predict how temporal structure in these sequences might affect their perceived relationships.
Initially we had intended discussing the rate of evolution, however, we did not have enough full genome sequencing data of BTV-8 to warrant/justify such a topic. From personal communication with our peers in the BTV field, an imminent publication will include genome data for over 100 full genome sequences of BTV-8 in France and will discuss the rate of evolution within.
One thing that would have been helpful, especially in re the discussion of wind and midge dispersal etc., is a more detailed map that clearly indicates the wind simulations and midge incursions you are proposing.
We have added forward trajectory maps for a selected number of potential incursions and sources as supplementary information (Figure S2). These demonstrate in more detail the potential for midges to be borne on the wind from sources in Syria, Lebanon and Israel

Reviewer 2 Report
The article by Rajko-Nenow et al is an epidemiological investigation of the origin of a BTV outbreak that occured in the Lakarna region in Cyprus in September 2016.
I enjoyed reading the article as it is well structured and pertains original results along with an original approach for the concise representation of phylogenetic relationships between isolates.
I only have a few minor comments:
- References are missing throughout the methods section (e.g. genome assembly)
- what mode was used on the sequencer (paired end? read length?)
- "next generation sequencing" is obsolete wording, the authors should consider replacing it with "high throughput sequencing"
- I understand that reference based assembly is standard in the field. However, I am curious whether a de novo based approach could potentially highlight more differences/rearrangements? Since this is a bit off scope, could the authors briefly comment on this aspect in the point-by-point response?
Author Response
Reviwer-2
The article by Rajko-Nenow et al is an epidemiological investigation of the origin of a BTV outbreak that occured in the Lakarna region in Cyprus in September 2016.
I enjoyed reading the article as it is well structured and pertains original results along with an original approach for the concise representation of phylogenetic relationships between isolates.
I only have a few minor comments:
- References are missing throughout the methods section (e.g. genome assembly)
We thank the reviewer for their consideration of our paper and for the suggestions/findings raised. We have therefore implemented the changes where necessary as indicated below:
Reference to the following programs: FASTQC, Trim Galore, BWA-MEM and the DiversiTools were added
Lines 160- 165 were modified to “A pre-alignment quality check was performed using the FASTQC program v0.11.8 [http://www.bioinformatics.babraham.ac.uk/projects/fastqc/] and the Trim Galore script [22] was used for quality and adapter trimming of FASTQ files along with removal of short sequences (<50bp). Subsequently, reads were mapped to a range of reference sequences using the BWA-MEM tool [23] and then the DiversiTools software [http://josephhughes.github.io/DiversiTools/] was used to generate the consensus sequence.
- what mode was used on the sequencer (paired end? read length?)
Line 156. Text changed from Libraries were prepared using the Nextera XT library preparation kit and sequencing was performed using MiSeq Reagent kit v2 (Illumina, USA) on the MiSeq benchtop sequencer. “
Line 156. Text changed to “Libraries were prepared using the Nextera XT library preparation kit and paired end read sequencing was performed using MiSeq Reagent kit v2 (Illumina, USA) on the MiSeq benchtop sequencer. “
- "next generation sequencing" is obsolete wording, the authors should consider replacing it with "high throughput sequencing"
Line 144, text changed from “next generation sequencing” to "high throughput sequencing”
- I understand that reference based assembly is standard in the field. However, I am curious whether a de novo based approach could potentially highlight more differences/rearrangements? Since this is a bit off scope, could the authors briefly comment on this aspect in the point-by-point response?
We have found previously that de novo assembly may lead to errors in BTV genome assembly if not used with caution e.g. generating two long segments by repeating a short fragment of the sequence. However, in saying this, de novo assembly can be particularly useful when assembling a novel BTV serotypes for which the closest relative shares <70% nucleotide identity.
